# Chiral Phosphine Catalyzed Allylic Alkylation of Benzylidene Succinimides with Morita–Baylis–Hillman Carbonates

**DOI:** 10.3390/molecules28062825

**Published:** 2023-03-21

**Authors:** Chang Liu, Jianwei Sun, Pengfei Li

**Affiliations:** 1Shenzhen Grubbs Institute, Department of Chemistry, Guangdong Provincial Key Laboratory of Catalysis, College of Science, Southern University of Science and Technology (SUSTech), Shenzhen 518055, China; 2Department of Chemistry, The Hong Kong University of Science and Technology, Clear Water Bay, Kowloon, Hong Kong SAR, China; 3Southern University of Science and Technology Guangming Advanced Research Institute, Southern University of Science and Technology, Shenzhen 518055, China

**Keywords:** alkylation, MBH carbonate, maleimide, phosphine, succinimide

## Abstract

Owing to their unique chemical properties, α-alkylidene succinimides generally act as versatile synthons in organic synthesis. Compared with well-established annulations, nucleophilic alkylations of α-alkylidene succinimides are very limited. Accordingly, an organocatalytic allylic alkylation of α-benzylidene succinimides with Morita–Baylis–Hillman (MBH) carbonates was established. In the presence of a chiral phosphine catalyst, α-benzylidene succinimides reacted smoothly with MBH carbonates under mild conditions to furnish a series of optical active succinimides in high yields and enantioselectivities. Different from the reported results, the organocatalytic enantioselective construction of pyrrolidine-2,5-dione frameworks bearing contiguous chiral tertiary carbon centers was achieved via this synthetic strategy. Scaling up the reaction indicated that it is a practical strategy for the organocatalytic enantioselective alkylation of α-alkylidene succinimides. A possible reaction mechanism was also proposed.

## 1. Introduction

The pyrrolidine-2,5-dione scaffold is the core unit of numerous bioactive compounds with anticonvulsant [1], antimycobacterial [2], and antidepressant properties [3]. The chemistry of succinimides is fascinating and is receiving much attention. Particularly, α-alkylidene succinimides bearing multiple electron-withdrawing groups and nucleophilic and electrophilic sites are versatile building blocks in diversified organic synthesis (Figure 1A). Unsurprisingly, α-alkylidene succinimides could be used as Michael acceptors to afford enantioenriched succinimide derivatives (Figure 1B) [4,5]. Notably, as shown in Figure 1C, α-alkylidene succinimides have also been successfully employed not only as C2-synthons, opening robust access to spiropyrrolizidines [6,7,8,9,10,11,12,13] and pyrones [14], but also as C3-synthons, making way for complex fused rings [15,16,17,18]. However, in stark contrast, reports on the catalytic nucleophilic alkylation of α-alkylidene succinimides (Figure 1D) that result in the formation of either maleimides or succinimide derivatives are very limited. Up to now, only a handful of reports have been disclosed on the organocatalytic enantioselective nucleophilic alkylation of α-alkylidene succinimides. In 2010, Jiang and Tan et al. reported a bicyclic guanidine-catalyzed Mannich-type allylic addition reaction of *N*-aryl alkylidene-succinimides with imines (Figure 1E) [19]. Importantly, the reactions of both 2-methylidene-*N*-aryl succinimides and *N*-aryl esterlidene-succinimides led to the formation of enantioenriched maleimides. Furthermore, the reaction of *N*-aryl benzylidene-succinimides enabled the formation of optically active succinimides. In 2016, Du’s group presented a bifunctional squaramide-catalyzed asymmetric Michael addition reaction of α-alkylidene succinimides with nitroalkenes to afford chiral functionalized succinimides (Figure 1F) [20]. Differently, as established by Wang and Yuan et al. in 2020, a bifunctional thiourea-catalyzed asymmetric Michael addition of benzylidene succinimides to β-trifluoromethyl enones followed by a 1,3-proton shift furnished a series of F_3_C-containing chiral Rauhut–Currier-type products (Figure 1G) [21]. Recently, Wan and Wang et al. developed a cinchona thiourea-catalyzed enantioselective Mannich reaction between benzothiazolimines and α-benzylidene succinimides for the synthesis of chiral benzothiazol succinimides (Figure 1H) [22].

Notably, Morita–Baylis–Hillman (MBH) carbonates were successfully employed as versatile allylic alkylation reagents to undergo asymmetric allylic substitutions with a wide range of nucleophiles with the aid of a chiral nucleophilic catalyst [23]. In 2012, Huang, Tan, and Jiang et al. reported a hydroquinidine-catalyzed allylic alkylation of Morita–Baylis–Hillman (MBH) carbonates with *N*-itaconimides as nucleophiles to afford a variety of multifunctional chiral α-methylene-β-maleimide esters (Figure 1I) [24]. However, over the past decade, the organocatalytic enantioselective nucleophilic alkylation of α-alkylidene succinimides with MBH carbonates as allylic alkylation reagents has remained silent [25,26]. Notably, as shown in Figure 1J, we successfully established an organocatalytic enantioselective [1 + 4]-annulation of MBH carbonates with a series of electron-deficient alkenes, such as β,γ-unsaturated α-keto esters and chalcones [27], *ortho*-quinone methides [28], α,β-unsaturated imines [29], 2-enoylpyridine *N*-oxides [30], and modified enones [31]. Particularly, we realized an organocatalytic regio- and enantioselective allylic alkylation of indolin-2-imines with MBH carbonates [32]. To fill this gap and as part of our ongoing interest in the field of organocatalytic asymmetric transformation of MBH carbonates, here we disclose a chiral phosphine-catalyzed allylic alkylation of α-benzylidene succinimides with MBH carbonates (Figure 1K). Different from the previous work, this strategy features nucleophilic phosphine catalysis and mild conditions without additives, and it works well over a broad substrate scope to furnish succinimide derivatives bearing contiguous chiral tertiary carbon centers in high yields with high asymmetric induction.

## 2. Results and Discussion

We started our investigation with the model reaction between 3-benzylidene-1-phenylpyrrolidine-2,5-dione **1a** and MBH carbonate **2a** in CH_2_Cl_2_ at room temperature for 16 h (Table 1). Initially, the reaction proceeded smoothly in the presence of phosphine **C1** to furnish the desired succinimide **3aa** at a 76% yield with 29% ee and 10:1 dr (Table 1, entry 1). To improve the efficiency and stereoselectivity, the chiral nucleophilic phosphine catalyst was carefully screened (Table 1, entries 2–8). Pleasingly, the **C3**-catalyzed reaction afforded the desired product **3aa** at an 85% yield with 56% ee and 10:1 dr (Table 1, entry 3). The use of **C4** as a catalyst enabled the formation of product **3aa** at a 91% yield with 62% ee and 8:1 dr (Table 1, entry 4). Particularly, the desired product **3aa** was obtained at a 95% yield with 85% ee and 11:1 dr when phosphine **C6** was employed (Table 1, entry 6). Further modifying the catalyst structure did not enhance the efficiency or the asymmetric induction (Table 1, entries 7 and 8). The effect of the substituent on the nitrogen atom of α-alkylidene succinimides was also surveyed. The **C6**-catalyzed reaction of 3-benzylidene-1-methylpyrrolidine-2,5-dione **1b** furnished the corresponding product **3ba** at a 94% yield with 77% ee and 9:1 dr (Table 1, entry 9). Notably, product **3ca** was obtained at a 90% yield with 92% ee and 10:1 dr from the **C6**-catalyzed reaction of 1-benzyl-3-benzylidenepyrrolidine-2,5-dione **1c** (Table 1, entry 10). With these encouraging data in hand, we then further optimized reaction conditions to obtain better results. The screening of reaction media disclosed that CH_2_Cl_2_ was suitable (Table 1, entries 11–14). The concentration was found to have a large effect on the reaction (Table 1, entries 15–18), enabling the formation of product **3ca** at a 92% yield with 92% ee and 11:1 dr (Table 1, entry 17). Neither increasing nor lowing the temperature could achieve better results (Table 1, entries 19–20). Prolonging reaction time to 36 h, the yield was increased to 96% without compromising stereoselectivity (Table 1, entry 21). However, further prolonging reaction time decreased the enantioselectivity (Table 1, entry 22). As a result, we identified the optimal reaction conditions: when 1-benzyl-3-benzylidenepyrrolidine-2,5-dione **1c** (0.05 mmol) was treated with MBH carbonate **2a** (0.06 mmol) in the presence of catalyst **C6** (10 mol%) in CH_2_Cl_2_ (0.75 mL) at room temperature for 36 h, the desired succinimide **3ca** was obtained at a 96% yield with 92% ee and 11:1 dr.

With the optimized reaction condition in hand, we then examined the substrate scope. As shown in Figure 2, the scope of α-benzylidene succinimide **1** was investigated with the **C6**-catalyzed reaction of MBH carbonate **2a**. Succinimides with different substitute groups (R^1^) on the nitrogen atom were tested. Under standard conditions, product **3aa** was obtained at a 96% yield with 81% ee and 11:1 dr, and **3ba** was obtained at a 91% yield with 81% ee and 5:1 dr, respectively. Furthermore, the succinimide with *n*-Bu group **3da** was isolated at a 91% yield with 82% ee and 11:1 dr. In particular, the **C6**-catalyzed reaction of 3-benzylidene-1-(*t*-butyl)pyrrolidine-2,5-dione **1e** generated the desired product **3ea** at a 94% yield with 92% ee and 14:1 dr. Moreover, the α-benzylidene succinimide with naphthalen-1-ylmethyl residue **1f** was also compatible to afford the corresponding product **3fa** at a 91% yield with 81% ee and 13:1 dr. The effect of aromatic group Ar^1^ was also surveyed. In general, a wide range of α-benzylidene succinimides **1g**–**n** reacted smoothly with MBH carbonate **2a** under standard conditions to give the corresponding succinimides **3ga**–**na** at high yields (86–94%) and stereoselectivities (76–98% ee, 9:1–14:1 dr). Both electron-donating (Me, MeS, and MeO) and electron-withdrawing (F, Cl, and Br) groups could be introduced into the aromatic ring of residue Ar^1^ with a slight effect on the efficiency and asymmetric induction. With one exception, the **C6**-catalyzed reaction of 1-benzyl-3-(2-methoxybenzylidene)pyrrolidine-2,5-dione **1k** resulted in the formation of product **3ka** at a 94% yield with 28% ee and 3:1 dr. It was found that 1-benzyl-3-(naphthalen-2-ylmethylene)pyrrolidine-2,5-dione **1o** was also tolerated to give the desired product **3oa** at a 93% yield with 85% ee and 11:1 dr.

Subsequently, the scope of MBH carbonates was also investigated with the **C6**-catalyzed reaction of α-benzylidene succinimide **1e** (Figure 3). To our delight, various MBH carbonates **2b**–**f** with different substituents (Ar^2^) were found to be compatible under standard conditions to afford the corresponding succinimides **3eb**–**ef** at a 90–94% yield with 92–95% ee and 10:1–15:1 dr. No significant electronic effect on the aromatic moiety was observed. Particularly, the MBH carbonate bearing thiophen-2-yl group **2g** reacted smoothly with α-benzylidene succinimide **1e** to afford the desired product **3eg** at a 91% yield with 92% ee and >20:1 dr. Moreover, the ester groups of MBH carbonate **2** had a slight effect on the efficiency and stereoselectivity, furnishing products **3eh**–**ek** at an 88–94% yield with 91–93% ee and 7:1–15:1 dr. Taken together, these encouraging results indicated that the chiral phosphine-catalyzed allylic alkylation of α-benzylidene succinimides with MBH carbonates was achieved, furnishing succinimide derivatives bearing contiguous chiral tertiary carbon centers at high yields with asymmetric induction.

To demonstrate the synthetic potential, the **C6**-catalyzed reaction was scaled up to 0.5 mmol of the starting material under standard reaction conditions. The corresponding product **3ea** was obtained at a 92% yield with 94% ee and 14:1 dr (Figure 4A). The absolute configuration of **3la** was unambiguously confirmed by X-ray crystallography (CCDC 2244711 (**3la**) contains the supplementary crystallographic data for this paper. These data can be obtained free of charge from The Cambridge Crystallographic Data Centre via www.ccdc.cam.ac.uk/data_request/cif. For details concerning the crystal structure of **3la** see the Appendix A as well). The stereochemistry of other products was assumed by analogy. On the basis of the reported results and our previous work [33], a proposed reaction mechanism was shown in Figure 4B. The nucleophilic attack of organocatalyst **C6** to MBH carbonate **2a** formed the intermediate **IM-I** and released the basic *t*-BuO^−^ ion to deprotonate α-benzylidene succinimide **1e** to afford the intermediate **IM-II**. The intermediate **IM-II** could isomerize into the intermediate **IM-III**. Via **IM**–**IV**, the reaction of intermediate **IM-I** with intermediate **IM-III** led to the formation of the desired product **3ea** and regenerated the catalyst **C6**.

## 3. Materials and Methods

All chemicals were used without further purification as commercially available unless otherwise noted. Thin-layer chromatography (TLC) was performed on silica gel plates (60F-254) using UV light (254 and 365 nm). Flash chromatography was conducted on silica gel (300–400 mesh). NMR (400, 500, or 600 MHz for ^1^H NMR; 100 or 126 MHz for ^13^C NMR; 376 MHz for ^19^F NMR) spectra were recorded in CDCl_3_ with TMS as the internal standard. Chemical shifts are reported in ppm, and coupling constants are given in Hz. Data for ^1^H NMR are recorded as follows: chemical shift (ppm), multiplicity (s, singlet; d, doublet; t, triplet; q, quartet; m, multiplet; dd, doublet–doublet), coupling constant (Hz), and integration. Data for ^13^C NMR are reported in terms of chemical shift (δ, ppm). Data for ^19^F NMR are reported in terms of chemical shift (δ, ppm). High-resolution mass spectral (HRMS) analyses were recorded on a Thermo Scientific Q Exactive Orbitrap mass spectrometer (Bremen, Germany) with ESI source. The crystal structure and data were recorded on a Rigaku HomeLab diffractometer. More details can be found in the Appendix A. In addition, the X-ray of **3la**, copies of NMR, and chiral HPLC analysis can be found in the Appendix A.

### 3.1. General Procedure for the Synthesis of α-Benzylidene Succinimide ***1***

Triphenylphosphine (10.5 mmol) was added to a solution of substituted 1-R^1^-1*H*-pyrrole-2,5-dione (10 mmol) and aldehyde (11 mmol) in EtOH (100 mL) at room temperature. The reaction mixture was stirred at room temperature overnight. When the reaction was completed (monitored by TLC), the reaction mixture was filtered, and the precipitation was washed with ethanol and dried to afford α-benzylidene succinimide **1**.

### 3.2. General Procedure for the Synthesis of MBH Carbonate ***2***

1,4-Diazabicyclo[2.2.2]octane (DABCO, 10.5 mmol) was added to a solution of aldehyde (10 mmol) in acrylate (20 mL) at room temperature. The reaction mixture was stirred at room temperature for 3–7 days. When the reaction was completed (monitored by TLC), the reaction mixture was purified by silica gel column chromatography to afford MBH alcohol.

4-(Dimethylamino)pyridine (DMAP, 2.08 mmol) was added to a solution of MBH alcohol and Boc-anhydride (15 mmol) in CH_2_Cl_2_ (30 mL) in batches. When the reaction was complete (monitored by TLC), the organic phase was washed with distilled water (2 × 20 mL) and dried over anhydrous Na_2_SO_4_, and the solvent was removed under reduced pressure. The residue was purified by silica gel column chromatography, affording MBH carbonate **2**.

### 3.3. General Procedure for the Phosphine-Catalyzed Allylic Alkylation

α-Benzylidene succinimide **1** (0.05 mmol), MBH carbonate **2** (0.06 mmol), and phosphine **C6** (10 mol%) were added to a solution of CH_2_Cl_2_ (0.75 mL). The reaction mixture was stirred at room temperature for 36 h. After the removal of the solvent, the crude residue was purified by preparative TLC (petroleum/ethyl acetate = 2:1) to obtain the desired product **3**.

### 3.4. Scale-Up of the Allylic Alkylation

MBH carbonate **2a** (175 mg, 0.6 mmol) and catalyst **C6** (25 mg, 0.05 mmol) were added to a solution of 3-benzylidene-1-(tert-butyl)pyrrolidine-2,5-dione **1e** (122 mg, 0.5 mmol) in CH_2_Cl_2_ (7.5 mL). The reaction mixture was stirred at room temperature for 36 h. Then, the mixture was purified by silica gel column chromatography (eluent: petroleum ether/EtOAc = 2:1) to yield the desired product **3ea** at a 92% yield (196 mg, 94% ee, 14:1 dr).

## 4. Conclusions

In conclusion, we developed an organocatalytic enantioselective allylic alkylation of α-benzylidene succinimides with MBH carbonates. With the aid of a chiral nucleophilic phosphine, a broad scope of α-benzylidene succinimides reacted smoothly with MBH carbonate to furnish functionalized α-benzylidene succinimides at high yields with high stereoselectivities. Importantly, this synthetic strategy not only achieved nucleophilic phosphine catalysis but also realized the asymmetric construction of enantioenriched succinimide derivatives bearing contiguous chiral tertiary carbon centers.

## Data Availability

The data presented in this study are available in the Appendix A.

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
