# Peer review of "Chiral Phosphine Catalyzed Allylic Alkylation of Benzylidene Succinimides with Morita–Baylis–Hillman Carbonates"

_molecules, 2023, doi:10.3390/molecules28062825_

Round 1

Reviewer 1 Report

In their manuscript Sun, Li and collaborator describe the stereoselective organocatalytic allylic alkylation of α-benzylidene succinimides with Morita-Baylis-Hillman (MBH) carbonates. The organocatalysts developed were chiral phosphine compounds bearing amino units in order to combine nucleophilic activation of the carbonate and basic activation of the nucleophile.

The enantioenriched chiral products were obtained in good yields, with high diastereoselective ratio and high ee%. The authors investigated into detail the scope of the reaction, the possible mechanism and applied the organocatalytic method to a large scale synthesis confirming the high selectivity properties.

The manuscript is very well written, with clear description of the objectives of the work. Results are of high interest for the development of stereoselective methods, all compounds were fully characterized, and all details are reported in the experimental section and in the supporting information file. Overall the manuscript is of certain interest for the readers of Molecules, therefore it deserves publication. I did not find any small mistake or comment on the paper that I consider suitable for publication as it is.

Author Response

Many thanks for your comments.

Reviewer 2 Report

1) in SI please add spaces when appropriate (typically in 2.33g and mp:100 phrases)

2) please in SI change "yields" to "yield"

3) Are compounds 2a-j known? If not, please provide MS or elemental analysis data?

4) please in SI change "racmic" to "racemic"

5) please in SI add degree sign to optical rotation angle values

6) dr values of compounds 3 are doubtful, i.e. from HPLC for 3aa: dr = (82.954+8.812):(0.610+7.624) = 8.324 but not 11:1 as postulated in the Manuscript. Please check values for other compounds. If dr values are provided for major enantiomer, please point it out in the Manuscript and change footnote d in Table 1 accordingly.  

Author Response

1) in SI please add spaces when appropriate (typically in 2.33g and mp:100 phrases)

Revision: Corrections have been made.

2) please in SI change "yields" to "yield"

Revision: Corrections have been made.

3) Are compounds 2a-j known? If not, please provide MS or elemental analysis data?

Revision: These compounds are known. Accordingly, we just gave the NMR analysis to make sure based on the reported data.

4) please in SI change "racmic" to "racemic"

Revision: Corrections have been made.

5) please in SI add degree sign to optical rotation angle values

Revision: Degree sign have been added.

6) dr values of compounds 3 are doubtful, i.e. from HPLC for 3aa: dr = (82.954+8.812):(0.610+7.624) = 8.324 but not 11:1 as postulated in the Manuscript. Please check values for other compounds. If dr values are provided for major enantiomer, please point it out in the Manuscript and change footnote d in Table 1 accordingly.  

Revision: The dr was determined by 1H NMR, which might be different from the data shown by HPLC. On the other hand, the dr value is provided for major enantiomer, which has been noted.

Reviewer 3 Report

In this manuscript entitled "Chiral phosphine catalyzed allylic alkylation of benzylidene succinimides with Morita-Baylis-Hillman carbonates' the authors have reported allylic alkylation of α-benzylidene succinimides with Morita-Baylis-Hillman (MBH) carbonates. The reaction has been systematically optimized to achieve high yields and high enantioselectivity. Authors also present very good substrate scopes with various succinimide and MBH substrates. 

There are minor issues that the authors need to be considered including in the revised version of the manuscript. With that this reviewer recommend the manuscript for publication in this esteemed journal after the following modifications:

1) Some important recent publications can be included in the introduction when describing the utilization of MBH carbonates in asymmetric allylic substitutions reactions (Org. Chem. Front., 2014, 1 1152-1156 and Org. Biomol. Chem., 2014,12, 5071-5076)

2) Very similar reaction mechanism is described in Org. Lett. 2021, 23, 5571−5575. Authors may consider citing this publication when describing the mechanism. 

3) In the MBH carbonates scope the authors have shown all aromatic substituents at the Ar2 position. Have authors investigated the aliphatic substituents or even no substitution at that position?    

Author Response

1) Some important recent publications can be included in the introduction when describing the utilization of MBH carbonates in asymmetric allylic substitutions reactions (Org. Chem. Front., 2014, 1 1152-1156 and Org. Biomol. Chem., 2014,12, 5071-5076)

Revision: Org. Biomol. Chem., 2014, 12, 5071-5076 was cited as ref 25.

         Org. Chem. Front., 2014, 1 1152-1156 was cited as ref 26.

2) Very similar reaction mechanism is described in Org. Lett. 2021, 23, 5571−5575. Authors may consider citing this publication when describing the mechanism. 

Revision: Org. Lett. 2021, 23, 5571−5575 was cited as ref 34.

3) In the MBH carbonates scope the authors have shown all aromatic substituents at the Ar2 position. Have authors investigated the aliphatic substituents or even no substitution at that position?    

Revision: We failed to obtain MBH carbonates with the aliphatic substituents, so no investigations of those MBH carbonates. The MBH carbonate with no substitution at that position, methyl 2-(((tert-butoxycarbonyl)oxy)methyl)acrylate, reacted with α‐benzylidene succinimide 1a under standard conditions to give the desired product in 86% yield, but the ee was poor, about 10%. Accordingly, this case was not given in the main text.

Reviewer 4 Report

In addition to the fact that the chemistry of succinimides is extremely interesting in itself, the relevance of the presented article lies in the fact that α-alkylidensuccinimides can also act as universal synthons in both organic and organoelement synthesis, and are widely used in the design of a number of biologically active compounds. Among which, drugs with antidepressant properties. The creation of the latter is especially important given the currently known complications caused by such a transferred disease as Covid_19.

The authors have developed original approaches to organocatalytic enantioselective allyl alkylation of various α-benzylidensuccinimides using Morita-Baylis-Hillman (MBH) carbonates with high yields and stereoselectivity as reagents. Using the proposed strategy, within the framework of the peer-reviewed study, it was possible to solve a non-trivial problem associated with the construction of enantio-enriched succinimide derivatives containing closely spaced chiral tertiary carbon centers.

The modern physico-chemical methods used in the work leave no doubt about the reliability of the results obtained

Author Response

Many thanks for your comments.